# Response of Chironomids (Diptera, Chironomidae) to Environmental Factors at Different Spatial Scales

**DOI:** 10.3390/insects15040272

**Published:** 2024-04-14

**Authors:** Bruno Rossaro, Laura Marziali

**Affiliations:** 1Department of Agricultural and Environmental Sciences (DISAA), University of Milan, Via Celoria 2, 20133 Milan, Italy; 2National Research Council—Water Research Institute (CNR-IRSA), Via del Mulino 19, 20861 Brugherio, Italy; laura.marziali@irsa.cnr.it

**Keywords:** aquatic insects, environmental factors, spatial factors

## Abstract

**Simple Summary:**

Chironomids are probably the most speciose family among aquatic insects, colonizing almost all freshwater habitats. The emphasis of the present research is on: (1) taxonomic composition as the most efficient tool for describing biodiversity in natural habitats; and (2) natural habitat type as the most important factor able to explain different chironomid assemblages. This is because the habitat type summarizes a combination of different biotic and abiotic factors present at each site, determining taxa assemblages. Different spatial scales are often proposed as relevant factors for modeling community composition, but with regard to chironomids, spatial factors act only at a small scale, while environmental factors are the most important determinants for species distribution.

**Abstract:**

Factors responsible for species distribution of benthic macroinvertebrates, including responses at different spatial scales, have been previously investigated. The aim of the present research was to review the most relevant factors explaining chironomid species distribution focusing on factors operating at different spatial scales, such as latitude, longitude, altitude, substrate, salinity, water temperature, current velocity, conductivity, acidity, dissolved oxygen, nutrient content etc. acting at regional levels and at a large or small water basin level. Data including chironomid species abundances from different lentic and lotic waters in Italy and other surrounding countries were analyzed using partial canonical correspondence analysis (pCCA) and multiple discriminant analysis (DISCR). Spatial analyses, including univariate Moran’s I correlograms, multivariate Mantel correlograms and Moran’s eigenvector maps (MEMs), were thereafter carried out. The results showed that habitat type, including different types of lotic waters (i.e., kryal, crenal, rhithral, potamal) and different lake types (i.e., littoral, sublittoral, profundal zones), is the most significant factor separating chironomid assemblages, while spatial factors act only as indirect influencers.

## 1. Introduction

Chironomids are amongst the most speciose families of aquatic insects, with about 15,000 described species, contributing to a large portion of the insect diversity in the benthos. Their ecological tolerance varies among species, with some being tolerant to extreme environmental conditions [1]. Various species have been shown to respond to a broad suite of different environmental (=abiotic) factors, such as water temperature, salinity and sediment composition [2], although biotic factors (competition, predation) are also important [3]. Furthermore, interactions between different combinations of abiotic and biotic factors as well as spatial and temporal variables are complex, making the species distributions difficult to interpret. Such complexities are further exacerbated in environmentally unstable aquatic habitats such as those in the Mediterranean [4], which are known to show broad hydro-morphological variations among years. The hydrological regime of those habitats will likely become even more unstable due to global climate change [5].

Historical biogeographic factors are known to influence chironomid species composition in aquatic habitats at large (continental) spatial scales [6], but in more specific habitats such as headwater streams, it can be supposed that environmental factors may act at a smaller spatial scale [7]. Despite this described complexity, often only a few key factors may account for the largest portion of observed variation [8]. The scarcity of samples and data available for study coupled with different sampling methods and tools and differing taxonomic levels used (i.e., family, genus, species), produces further uncertainty in interpreting results. Many studies considered responses at different spatial scales [9], but few studies evaluated the responses of chironomids at different spatial scales [10], and they were primarily focused on small spatial scales [11,12].

The aim of the present study is to confirm the relevance of environmental factors acting at small spatial scales and to assess the potential importance of factors acting at larger spatial scales by using traditional multivariate methods, such as partial canonical correspondence analysis (pCCA) and multiple discriminant analysis (DISCR), and more recent tools used to analyze spatial data based on Moran’s eigenvector maps (MEMs) [13].

## 2. Materials and Methods

The input data matrix included 788 study sites in 11 different habitat types (Table 1) encompassing lotic and lentic water bodies; the former were classified according to Illies and Botosaneanu [14], and the latter according to Buraschi et al. [15] and Tartari et al. [16].

The running water types were separated into kryal, crenal, rhithral and potamal zones, adding the kryal zone (glacial streams) to the Illies classification [14]. The lake classification following [15,16] separates lakes with a prevalent profundal zone (AL03), from littoral (AL04), littoral–sublittoral (AL05) and prevalent sublittoral (AL06) lakes.

The samples were collected with different sampling techniques according to habitat type. Kick sampling was used in kryal, crenal and rhithral habitats, drift nets were used in potamal stretches of rivers, PONAR grabs were used in different lake types, Ekman grabs were used in Alpine lakes. Species abundances were standardized to 1 m^2^ bottom surface to allow comparison between samples collected in different habitat. Samples were collected mostly in Italy [2,8], and secondarily in other countries: Austria, Germany [17], Switzerland [18,19], Montenegro [20], Greece [21], Algeria [22,23] and France (Garonna River). The maps with sampling sites were prepared using QGIS version 3.34.1 Prizren (2009) (https://www.qgis.org/it/site/, accessed on 1 February 2024) [24].

Raw data comprised 19,531 samples collected in 788 study sites with 255 variables, i.e., 22 environmental factors/variables and 233 chironomid species. Only samples including values of at least ten variables and variables present in at least 100 samples were considered for analysis. This selection left 782 sites with 101 species and nine environmental variables: altitude (altit) (m a.s.l.), distance from the source (dist) (km), conductivity (cond) (µS cm^−1^), dissolved oxygen (O_2_) (mg L^−1^), oxygen saturation (O_2_%) (% saturation), water temperature (temp) (°C), pH, total phosphorous (TP) (µg L^−1^) and ammonium hydroxide (NH_4_-N) (mg L^−1^). Latitude and longitude expressed in the coordinate reference system WGS 84/UTM zone 32, three spatial scales (regional, large and small water basin), habitat type, year and month were included as factors in the database. Mean values of the variables calculated for each of the 39 large water basins are provided in Appendix A.

To analyze factors responsible for species distribution in the presence of potential influence of spatial factors, data were analyzed with a pCCA after log(x + 1) transformation of the species matrix and standardization of environmental variables. The pCCA was carried out with the species matrix as dependent variables, the nine environmental variables as constraining variables (independent variables) and spatial coordinates as conditioning variables (covariates). Forward and backward selection of environmental variables was performed to select the variables accounting for the largest source of variation.

To confirm the results of pCCA, an inverse pCCA (pCCAi) was calculated, with spatial variables as constraining factors and environmental variables as conditioning factors; in other words, constraining and conditioning factors were exchanged.

To confirm the importance of habitat type in driving species composition, a DISCR analysis was performed with species as dependent variables and habitat type as a discriminant factor. Data were analyzed using the R-package ‘vegan’ [25].

To analyze the influence of spatial factors, the sites having the same spatial coordinates, but sampled on more than one date, were pooled into a single record, leaving a matrix with 264 sites, with 54 species present in at least 50 sites.

The analysis of spatial factors is needed because both environmental variables and species abundances can be influenced by values of the same variable measured at the surrounding sites. A correlogram measures this influence at increasing spatial scale. Univariate Moran’s I spatial correlograms were calculated to analyze the response of each environmental variable and of each species to increasing spatial scale. Multivariate Mantel’s correlograms were also calculated to analyze the spatial influence on the full set of environmental variables and the full set of species [25]. Correlograms were analyzed using the R-packages ‘vegan’ and ‘spdep’ [26].

To further examine the species response at different spatial scales, MEMs were calculated using the R function *mem*, which transforms the distance matrix into an eigenvector matrix, with sites as rows and eigenvectors as columns; the eigenvectors represent the spatial structure at decreasing spatial scales [26]. The species matrix was submitted to a multiscale pattern analysis (MSPA) [27] using the R function *mspa*, to recalculate the spatial structure of the data and to identify the scales of spatial variation more correlated with species. Calculations were performed using the R-packages ‘adespatial’, ‘spdep’ and ‘vegan’. As a further step, a spatial weighting matrix (SWM) was calculated using the R functions *mem.select*, *listw.candidates* and *listw.select* [28] to further select the subset of MEM eigenvectors which yields the highest correlation (i.e., the highest adjusted R^2^) with species matrix. As a last step, a canonical analysis (=redundancy analysis using the function *pcaiv* in the package ‘ade4’) was used to explain the structure of the species matrix constrained by MEM spatial variables selected starting from the best SWM (*sel.lw$best*). See Jombart et al. [27] for a detailed explanation of all of these steps.

Details of MEM, MSPA and SWM calculations are can be found in Bauman et al. [28] and Dray [29] and at the web site: https://cran.r-project.org/web/packages/adespatial/vignettes/tutorial.html (accessed on 1 February 2024).

The last step in data analysis was the calculation of diversity using the R-package ‘vegan’. To estimate diversity, three spatial scales (i.e., region, large and small water basin) were considered. The mean species number present in each area (=alpha diversity), calculated as the mean of values for each site within the area, the total number of species present in the area (=gamma diversity) and the ratio between gamma and alpha diversity (=beta diversity) were then calculated.

## 3. Results

The high number of species found (233) was expected, because of the diverse habitat analyzed, including both running waters and lakes from different countries. The list of species included in the data analysis is provided in Table 2, with the number of samples for each species.

The response of species to environmental factors, removing the potential confounding effect of spatial variables, was analyzed using pCCA. The nine environmental variables included as constraining variables accounted for 11% of total inertia, while spatial variables included as covariates explained less than 5% (Table 3a).

The inverse pCCAi, with spatial variables as constraining variables and environmental ones as conditioning variables was carried out to establish if spatial variables can have a direct influence on species response, having removed the possible influence of environmental variables. In pCCAi, spatial variables accounted for an even lower proportion of inertia, less than 4% (Table 3b).

The first eigenvalue accounted for 5.5%, and the second eigenvalue for 2.6% of variation in the pCCA (Table 4a) but only 1.1% and <1%, respectively, in the pCCAi (Table 4b); the other eigenvalues accounted for negligible proportions in both analyses.

Results of direct pCCA show a clear separation of high-altitude sites with low water temperature, from lowland sites (Figure 1a). The separation was evident along the first axis and was explained by an altitudinal temperature gradient. The second axis emphasized a clear trophic gradient, with sites rich in total phosphorus and poor in oxygen concentration separated from sites with low total phosphorous and high oxygen concentration.

The plot of sites evidenced a clear habitat separation (Figure 1b), with kryal sites grouped to the left and potamal sites at the bottom right. Large and small lakes (AL03–AL06) prevailed on the right. The alpine lakes (ALAlps) at high altitude were plotted near the kryal sites. The separation of lowland lakes (ME04 and ME07) from rhithral sites was less evident, even if rhithral sites were grouped in the center of the graph, while lakes surrounded them and potamal sites were well separated.

Species distribution in the plot (Figure 1c) mirrored that of sites (Figure 1b), with taxa characteristic of kryal and alpine lakes plotted on the left, species characterizing potamal and rhithral plotted on the right. The separation of species preferring crenal habitats or lakes was less evident here.

The pCCAi emphasized that the second axis separated sites and species according to latitude (Y) and longitude (X) (Figure 1d). The axes describing smaller spatial scales are all plotted in the left part of the graph. No clear separation of species according to their preferences is visible; only Tanytarsini are separated better than the species belonging to other tribes, but the species ordination did not separate groups with different ecological needs.

DISCR analysis using habitat types as factors showed a very good agreement between expected and observed classification (Figure 2, Table 5 and Appendix A). Some habitats such as AL03 and rhithral had more predicted sites than initially assigned, while others such as lakes (AL05–AL06) and potamal had a lower number of predicted sites than assigned, but in general the agreement was very good. These results confirmed direct pCCA results, i.e., that habitat type was a good predictor of chironomid assemblages.

The response of each environmental variable and of each species to increasing spatial scale, analyzed with univariate spatial correlograms, showed that spatial autocorrelation decreased rapidly with increasing distance class for all variables. The autocorrelations were generally low, below 1. The highest autocorrelation of environmental variables was observed for conductivity (Figure 3a), the highest autocorrelation of species was observed for *T. gregarius*, which appeared to be one of a few species with autocorrelations observable also at larger distances, but it was an exception (Figure 3b). Low values of autocorrelation mean that the value of a variable at a given site is only slightly related to the values in nearby sites and a still lower influence is observed at increasing spatial scales.

This result was confirmed examining the Mantel correlograms, which showed that Mantel’s autocorrelation was very low and vanished just after the first distance class; this was evident both for environmental variables (Figure 3c), and for chironomid assemblages (Figure 3d), confirming the absence of spatial autocorrelations except at the lowest distance.

The MEMs generated from SWM were calculated and numbered from 1 to 264, with 264 being the number of sites; MEM1 is the eigenvector map associated with the highest eigenvalue and maps the largest distances, MEM264 is the one associated with the lowest eigenvalue and maps the shortest distances. The full matrix of MEMs is given in Appendix A. The most significant MEMs were plotted in European maps; the eigenvectors were divided into five classes and are represented by different colors (Figure 4).

The MEM of the first axis was related to latitude, the MEM of the second and third to longitude, while the other axes could not be associated to a well-defined factor. Few significant correlations between environmental variables and MEM eigenvectors were observed (see Appendix A). The R^2^ correlations between the species matrix and MEM spatial predictors, calculated from the best SWM (see Data analysis), were filed in decreasing order (Table 6). In Appendix A, the completed list of R^2^ correlations calculated with two different methods (mem.gab.sel from *mem.select* and sel.lw$best from *listw.select*, see Section 2) are given [27]. 

Finally, redundancy analysis (RDA) was carried out to relate the sites x species matrix with the MEM eigenvector matrix. Detailed results of the RDA can be examined in Appendix A, where the scores of all species, sites and MEM variables are given. A summary of results is shown in Figure 5 and Figure 6. The separation of sites is still evident according to habitat type (Figure 5); in this case, the lakes are separated and plotted on the left, even if large, small and volcanic lakes are not well separated, kryal and alpine lakes are in the upper part of figure, while rhithral sites are on the right in the center, and potamal sites are at the bottom right. The separation according to habitat is in agreement with pCCA and DISCR analysis, even if the position of each habitat is changed. The species separation is consequent, with the species with highest spatial autocorrelation as *T. gregarius*, and *P. H. choreus* have the highest scores in RDA analysis (Figure 6a, Appendix A). Species preferring lentic habitat are plotted on the right, cold stenothermal species at the top and species characteristic of lotic habitats at the bottom right. The MEM variables are also plotted (Figure 6b); those with the highest R^2^ values are the ones most distant from the center, as is clearly seen when comparing Table 6 with Figure 6b.

Diversity for different habitats and for different spatial scales was then calculated to support the previous conclusions. The highest gamma diversity was observed in rhithral (95 species) and potamal (87) sites, and the lowest in volcanic lakes (28), while the highest alpha diversity was observed in lakes AL06 (16) and the lowest in alpine lakes (ALAlps) (13). The highest beta diversity was observed in rhithral (6) habitats and the lowest in volcanic lakes (2). A classification based on water basins shows that the highest gamma diversity (93) was observed in the Sarca basin, and the highest beta diversity in Adda (7); at a lower spatial scale, the highest gamma diversity (77) was observed in Lambro stream, and the highest beta diversity (5) in Bormida stream (see Appendix A for detailed results). The gamma and beta diversity decreased from the largest to the smallest spatial scale, respectively measuring 67 and 4.6 at the largest scale (regional), 47 and 3.2 at the intermediate scale (large water basin) and 26 and 1.8 at the smallest scale (small water basin).

## 4. Discussion

Our study showed that environmental variables such as substrate, salinity, water temperature, current velocity, conductivity, acidity, dissolved oxygen, nutrient content, etc., can be summarized in a single factor, i.e., habitat type, classified as kryal, crenal, rhithral, potamal zones of running waters, littoral, sublittoral, and profundal zones of lakes, summarizing preferences for different factors interacting with each other. Habitat type is able to explain the chironomid species composition, while geographic (i.e., latitude, longitude, altitude) and spatial factors (i.e., source distance, water depth) have only an indirect influence, corroborating results of previous studies [2,8]. In fact, pCCA emphasized that the highest proportion of inertia was accounted for by environmental variables summarized in the habitat type (Figure 1b), while spatial factors (i.e., latitude, longitude and their polynomial expansions) accounted for a much smaller proportion of inertia. DISCR analysis of the species matrix carried out using habitat as the discriminant factor confirmed that there was very good agreement between the ‘a priori’ and ‘a posteriori’ classification based on habitat type.

The hypothesis that spatial factors at different scales could significantly influence chironomid response was tested with spatial correlograms and with Moran’s maps. At small scales, spatial autocorrelation was evident only at the shortest distances, with the possible exception of few species such as *T. gregarius*, which also showed a moderate autocorrelation at larger distances, but for most species it was negligible also at the shortest distances. Considering large spatial scales, the Moran’s maps evidenced a spatial separation of species according to latitude and longitude, but this was interpreted as an artifact, because the sites at the highest altitudes with lower temperature are more frequent in the western part of the sampled area (Western Alps), suggesting a misleading influence of longitude, and the sites at lower latitude suffer an indirect effect of temperature, having higher temperatures [30,31] than the northern sites.

As regards diversity [32], in previous studies, the influence of spatial factors on chironomid diversity [11,12] was analyzed considering the effect of river order. The highest diversity of chironomids was found for streams of intermediate order, suggesting that factors acting at intermediate spatial scales are more relevant than factors at large and small spatial scales, in agreement with the old intermediate disturbance hypothesis [33]. We did not test influence of river order on diversity; our data emphasized that beta and gamma diversity decreased at the three different spatial scales (regional, large and small water basin), but this was expected given the decreasing extent of the geographic areas. We also observed the highest gamma and beta diversity in rhithral and potamal sites, and the lowest in kryal habitats, again confirming the importance of habitat in explaining chironomid distribution.

Taxonomic composition was considered too cumbersome in ecological studies in recent years by many authors, who tried to substitute or augment taxonomic composition with functional composition descriptors [34,35] or with species trait analysis [36,37]. The choice between taxonomic, functional and trait composition influenced the interpretation of results at different spatial scales. For example, it was emphasized [5] that species trait composition could be less affected by regional scale than taxonomic composition, while ecoregion and season accounted for 20.5% of the variance in functional composition, but only 10.9% of the variance in taxonomic structure [10]. In the present case, we considered only taxonomic composition, because in our opinion, trait analysis and functional descriptors are not sufficiently developed to be applied to chironomid species [37].

There are few or no publications that can be compared with our study in considering response of chironomids to different spatial scales. Feld & Hering [10] investigated benthic macroinvertebrates at different spatial scales (ecoregion, catchment, reach, site) in four different countries (Sweden, the Netherlands, Germany and Poland). Mykrä et al. [38] focused their study on the Fennoscandia bioregion with pCCA, and concluded that local factors are prevalent in explaining macroinvertebrate distribution, although regional factors are also relevant at a larger spatial scale. In the present analysis, the area sampled included Italy, France, Switzerland, Austria, Germany, Greece, Montenegro and Algeria, although samples from Italy were largely prevalent and included Western, Central, Eastern Alps, Prealps, lowland Po River, different rivers in Central–South Italy, large and small prealpine lakes and volcanic lakes (Appendix A), and the whole area was analyzed at three different spatial scales (regional, large and small water basin). Similar analyses (i.e., pCCA) were used [10,38] as in the current study; however, for various reasons, a direct comparison with our results is complicated. For instance, in those studies, only running water samples were analyzed, while we included both lotic and lentic samples. Furthermore, although both the current and previous studies included species-level data, in the current study, we focused only on chironomid assemblages, while in the studies of Feld & Hering [10] and Mykrä et al. [38], the whole benthic macrofauna was analyzed. Finally, fewer environmental and spatial variables were included in the current study (nine environmental and two spatial variables) compared to the previous studies: in [10], 31 and in [38], 15 environmental variables at four [10] (i.e., mega, macro, meso, micro), or three [38] (i.e., bioregion, ecoregion, drainage) hierarchical extents. Feld & Hering [10] concluded that interactions at different spatial scales confounded the interpretation of the results, but the meso-scale variables accounted for the largest source of variation; Mykrä et al. [38] considered local factors more important.

We can conclude that spatial scales are not relevant in modeling chironomid taxa assemblages. It is well known that at a very large spatial scale, considering wide global areas (Palaearctic, Nearctic, Oriental, Neotropic, Australian), chironomid species composition is quite different [6], but within the restricted area considered (i.e., the Mediterranean–Alpine area), the spatial factors have no influence, or they are mediated by other factors (e.g., altitude, water temperature etc.). Anthropogenic stress was not a focus of the present research [39], but even if many altered sites were included in the dataset, they did not substantially alter our conclusions about the prominent effect of habitat, although the well-known response of chironomids to oxygen shortage and eutrophication (measured as TP) [2,8,18] was confirmed here (Figure 1a).

## 5. Conclusions

Considering the response of benthic macroinvertebrates, and of chironomids in particular, to environmental factors, one concern is that important predictors may be missing in the model. This issue can be addressed by considering a more general factor such as habitat type, classified as littoral, sublittoral and profundal zone for lakes, and kryal, crenal, rhithral and potamal zone for running waters, as the best predictor of benthic macroinvertebrates in general and of chironomids in particular. Habitat type summarizes a suite of different environmental variables, such as substrate, salinity, water temperature, current velocity, conductivity, acidity, dissolved oxygen, nutrient content etc., avoiding the risk that some important predictors may be overlooked. The present database was analyzed with multivariate methods (pCCA, DISCR) able to combine different variables into few descriptors, i.e., eigenvectors with high inertia; we found that the eigenvectors with the highest inertia could be easily associated with habitat types. This was also confirmed by the analysis of spatial autocorrelation and Moran’s I eigenvector maps, which supported the conclusion that spatial factors act only as indirect drivers determining species composition, at least within an area with an extension comparable to the Mediterranean region.

## Figures and Tables

**Figure 1 insects-15-00272-f001:**
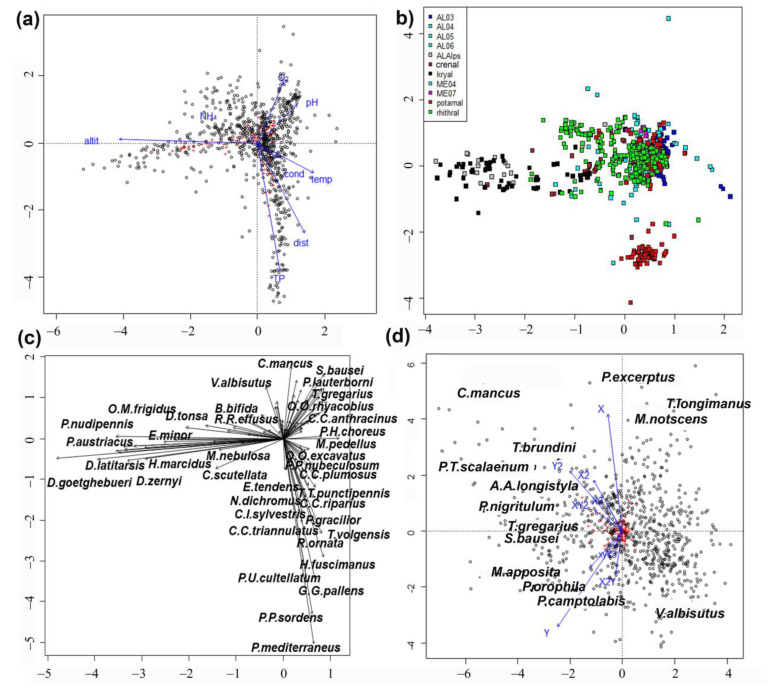
(**a**) pCCA results. Plot of environmental variables on the first two axes. Abbreviations: altitude (altit), distance from the source (dist), conductivity (cond), dissolved oxygen (O_2_), water temperature (temp), pH, total phosphorous (TP), ammonium hydroxide (NH_4_). (**b**) pCCA results. Plot of sites on the first two axes. Sites are grouped according to habitat and plotted with a different color. Abbreviations are as in Table 1. (**c**) pCCA results. Plot of species on the first two axes. Species abbreviations are as in Table 2. (**d**) pCCAi results: plot of spatial factors as constraining variables, with environmental variables as conditioning variables, X = longitude, Y = latitude. Species abbreviations are given in Table 2.

**Figure 2 insects-15-00272-f002:**
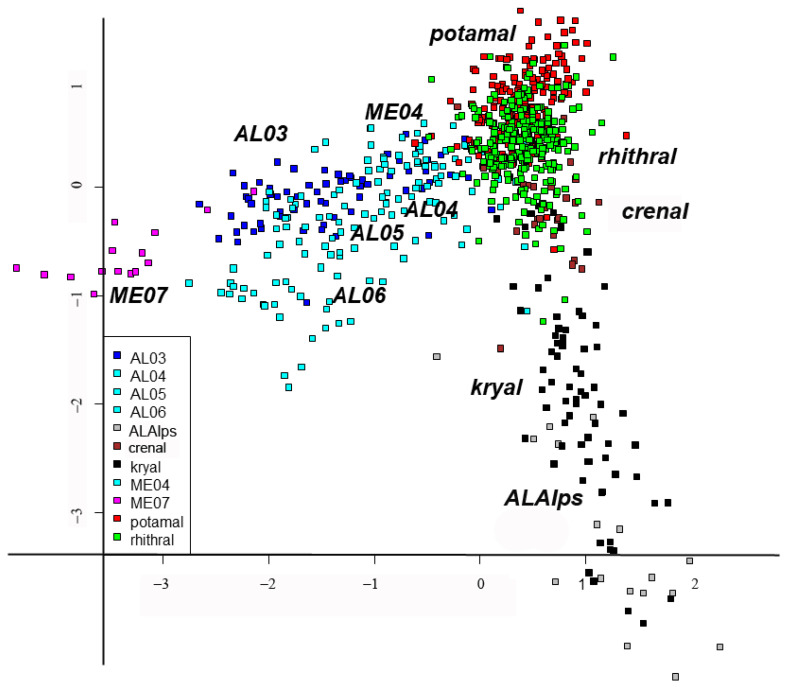
Discriminant analysis: plot of sites using habitat as a discriminant factor; different habitats are marked with different colors. Habitat type abbreviations are given in Table 1.

**Figure 3 insects-15-00272-f003:**
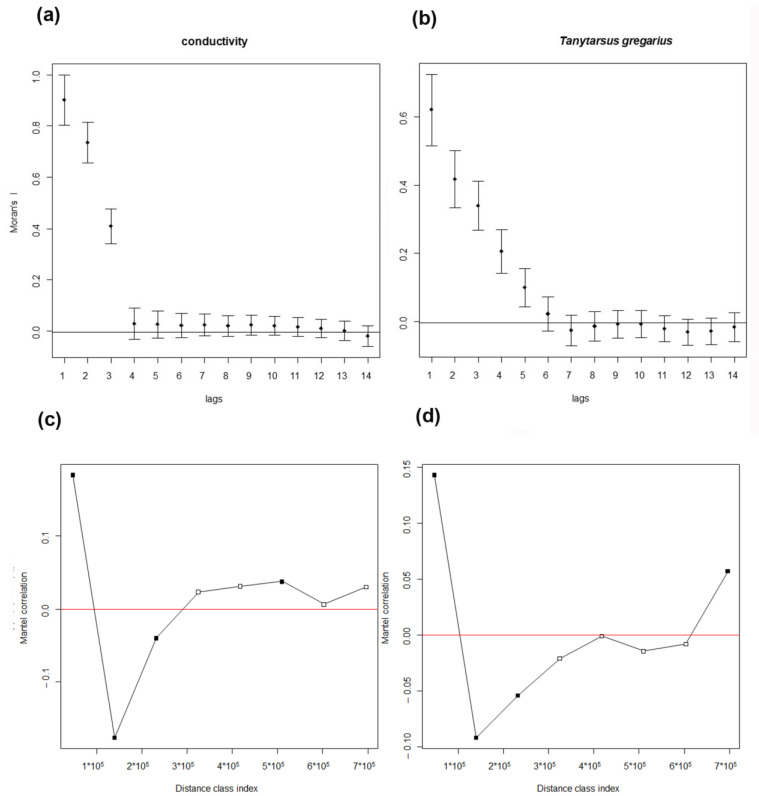
Univariate correlogram of conductivity (**a**) and of *T. gregarius* (**b**). Multivariate Mantel’s correlogram of environmental variables (**c**) and of species (**d**). Black squares mean significant values.

**Figure 4 insects-15-00272-f004:**
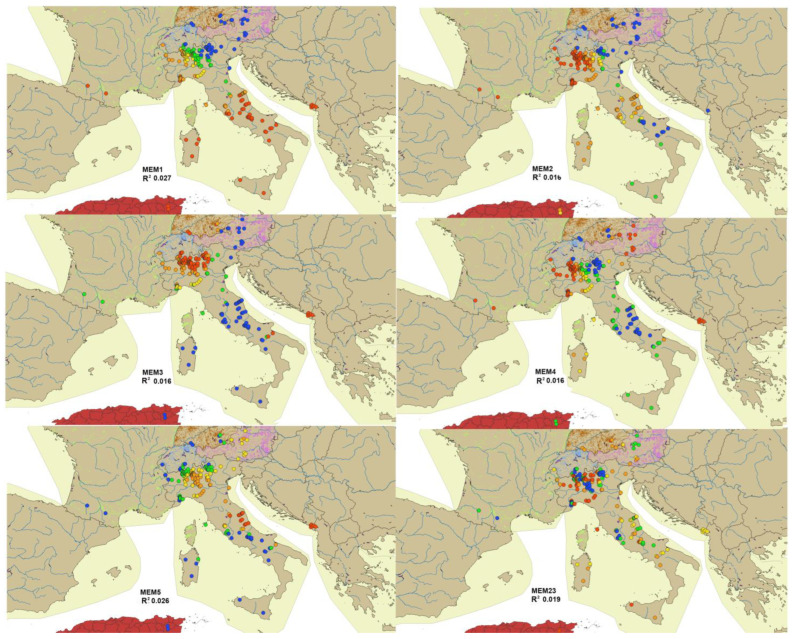
Plot of six (MEM1, MEM2, MEM3, MEM4, MEM5, MEM23) of the out of ten most significant Moran’s I eigenvectors, given in Table 6 with their R^2^ values. Values are grouped into five classes, with different colors (blue, green, yellow, orange, red) from highest to lowest; the full MEM matrix is given in Appendix A.

**Figure 5 insects-15-00272-f005:**
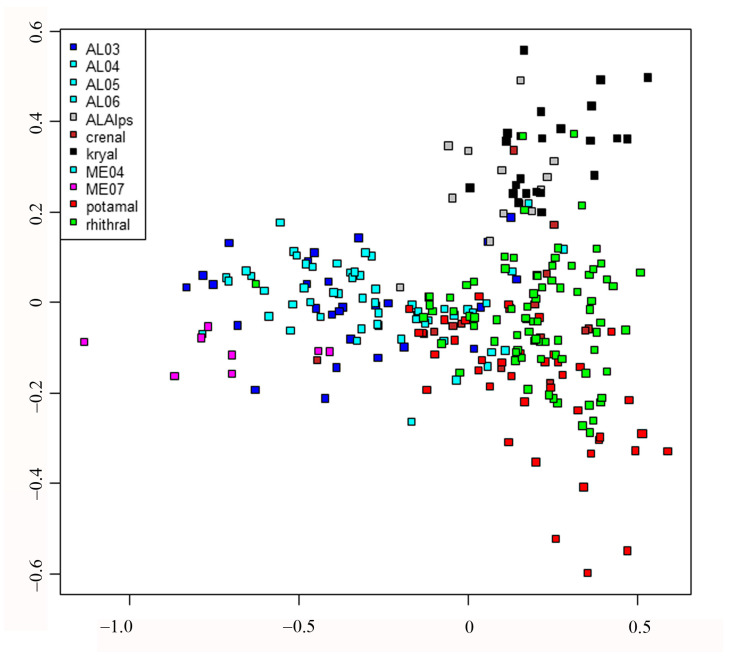
RDA results carried out using the species matrix constrained by spatial variables: map of sites grouped according to habitat, plotted with different colors. Habitat abbreviations are given in Table 1.

**Figure 6 insects-15-00272-f006:**
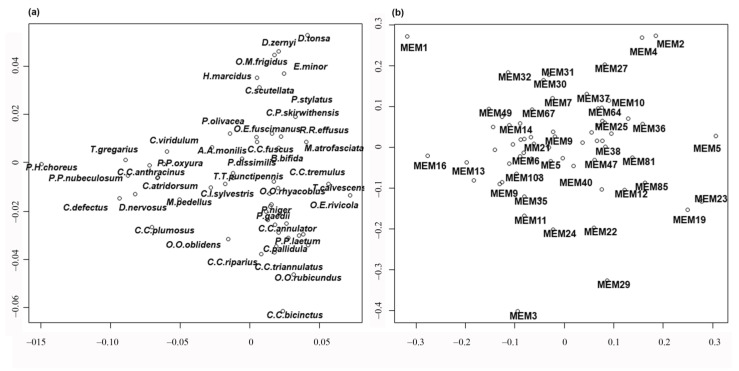
RDA results carried out using the species matrix constrained by spatial variables. (**a**) Plot of species. Species abbreviations are given in Table 2. (**b**) Plot of MEM variables.

**Table 1 insects-15-00272-t001:** Brief description of the habitat types sampled, with abbreviations used in figures (codes) and the number of sites sampled. See [15,16] for more details.

Water Bodies	Code	Habitat Description	Number of Sites
Lakes	AL03	Large lakes with 100 km^2^ surface, maximum depth above 120 m, below 800 m a.s.l. altitude, including littoral, sublittoral and profundal zones	68
	AL04	Lakes with maximum depth < 15 m, below 800 m a.s.l. altitude, polymictic, without a clear thermal stratification	8
	AL05	Lakes similar to AL04, but with thermal stratification, with littoral, sublittoral and profundal zone	53
	AL06	Lakes similar to AL05, but with depth above 15 m, with littoral, sublittoral and profundal zone	53
	ALAlps	Small alpine lakes, including types AL01, AL02, AL07, AL08, AL09, AL10 described in [15,16], all above 800 m a.s.l., with a calcareous or siliceous substrate	18
	ME04	Lakes collected in the Mediterranean area, except for the volcanic lakes ME07	22
	ME07	Volcanic lakes in Central Italy	15
Streams and rivers	Crenal	Springs	45
	Kryal	Glacial streams or cold springs near glacial streams	70
	Rhithral	Wadable streams	259
	Potamal	Large, not wadable rivers	177

**Table 2 insects-15-00272-t002:** List of species included in data analysis, in phylogenetic order according to the GBIF dataset (https://www.gbif.org/dataset/90d9e8a6-0ce1-472d-b682-3451095dbc5a, accessed on 1 February 2024), with abbreviations used in figures and the number of samples for each species (n).

Abbreviations	Species	Author	n
T. T. punctipennis	*Tanypus (Tanypus) punctipennis*	Meigen, 1818	83
P. H. choreus	*Procladius (Holotanypus) choreus*	(Meigen, 1804)	255
M. nebulosa	*Macropelopia nebulosi*	(Meigen, 1804)	68
A. A. longistyla	*Ablabesmyia (Ablabesmyia) longistyla*	Fittkau, 1962	68
A. A. monilis	*Ablabesmyia (Ablabesmyia) monilis*	(Linnaeus, 1758)	94
T. longimanus	*Trissopelopia longimanus*	(Stäger, 1839)	31
Z. barbatipes	*Zavrelimyia barbatipes*	(Kieffer, 1911)	63
C. pallidula	*Conchapelopia pallidula*	(Meigen, 1818)	349
R. ornata	*Rheopelopia ornate*	(Meigen, 1838)	73
P. branickii	*Pseudodiamesa branickii*	(Nowicki, 1873)	72
D. bertrami	*Diamesa bertrami*	Edwards, 1935	58
D. goetghebueri	*Diamesa goetghebueri*	Pagast, 1947	21
D. latitarsis	*Diamesa latitarsis*	(Goetghebuer, 1921)	48
D. cinerella	*Diamesa cinerella*	Meigen in Gistl, 1835	29
D. tonsa	*Diamesa tonsa*	(Walker, 1856)	184
D. zernyi	*Diamesa zernyi*	Edwards, 1933	85
S. spinifera	*Sympotthastia spinifera*	Serra-Tosio, 1968	61
P. gaedii	*Potthastia gaedii*	(Meigen, 1838)	90
P. longimanus	*Potthastia longimanus*	(Kieffer, 1922)	52
P. olivacea	*Prodiamesa olivacea*	(Meigen, 1818)	173
B. bifida	*Brillia bifida*	(Kieffer, 1909)	152
B. longifurca	*Brillia longifurca*	Kieffer, 1921	53
T. calvescens	*Tvetenia calvescens*	(Edwards, 1929)	310
E. ilkleyensis	*Eukiefferiella ilkleyensis*	(Edwards, 1929)	109
E. minor	*Eukiefferiella minor*	(Edwards, 1929)	124
E. brevicalcar	*Eukiefferiella brevicalcar*	(Kieffer, 1911)	57
E. claripennis	*Eukiefferiella claripennis*	(Lundbeck, 1898)	241
P. P. sordidellus	*Psectrocladius (Psectrocladius) sordidellus*	(Zetterstedt, 1838)	53
P. P. oxyura	*Psectrocladius (Psectrocladius) oxyura*	Langton, 1985	120
R. P. chalybeatus	*Rheocricotopus (Psilocricotopus) chalybeatus*	(Edwards, 1929)	136
R. R. effusus	*Rheocricotopus (Rheocricotopus) effusus*	(Walker, 1856)	121
R. R. fuscipes	*Rheocricotopus (Rheocricotopus) fuscipes*	Kieffer, 1909	210
P. niger	*Paracricotopus niger*	(Kieffer, 1913)	95
N. dichromus	*Nanocladius dichromus*	(Kieffer, 1906)	89
P. nudipennis	*Parorthocladius nudipennis*	(Kieffer, 1908)	42
S. semivirens	*Synorthocladius semivirens*	(Kieffer, 1909)	244
O. E. fuscimanus	*Orthocladius (Eudactylocladius) fuscimanus*	(Kieffer in K. & Thien., 1908)	82
O. E. rivicola	*Orthocladius (Euorthocladius) rivicola*	Kieffer, 1921	294
O. M. frigidus	*Orthocladius (Mesorthocladius) frigidus*	(Zetterstedt, 1838)	147
O. O. excavatus	*Orthocladius (Orthocladius) excavatus*	Brundin, 1947	179
O. O. oblidens	*Orthocladius (Orthocladius) oblidens*	(Walker, 1856)	184
O. O. rhyacobius	*Orthocladius (Orthocladius) rhyacobius*	Kieffer, 1911	164
O. O. rubicundus	*Orthocladius (Orthocladius) rubicundus*	(Meigen, 1818)	217
C. P. skirwithensis	*Cricotopus (Paratrichocladius) skirwithensis*	(Edwards, 1929)	100
C. P. rufiventris	*Cricotopus (Paratrichocladius) rufiventris*	(Meigen, 1830)	304
C. C. fuscus	*Cricotopus (Cricotopus) fuscus*	(Kieffer, 1909)	80
C. C. tremulus	*Cricotopus (Cricotopus) tremulus*	(Linnaeus, 1756)	179
C. C. annulator	*Cricotopus (Cricotopus) annulator*	Goetghebuer, 1927	193
C. C. triannulatus	*Cricotopus (Cricotopus) triannulatus*	(Macquart, 1826)	138
C. C. bicinctus	*Cricotopus (Cricotopus) bicinctus*	(Meigen, 1818)	326
C. C. trifascia	*Cricotopus (Cricotopus) trifascia*	Edwards, 1929	97
C. I. sylvestris	*Cricotopus (Isocladius) sylvestris*	(Fabricius, 1794)	218
C. I. intersectus	*Cricotopus (Isocladius) intersectus*	(Stäger, 1839)	33
C. dentiforceps	*Chaetocladius dentiforceps*	(Edwards, 1929)	46
P. excerptus	*Paratrissocladius excerptus*	(Walker, 1856)	21
H. marcidus	*Heterotrissocladius marcidus*	(Walker, 1856)	64
P. stylatus	*Parametriocnemus stylatus*	(Kieffer, 1924)	181
P. bathophila	*Parakiefferiella bathophila*	(Kieffer, 1912)	52
H. serratosioi	*Heleniella serratosioi*	Ringe, 1976	45
C. lobata	*Corynoneura lobata*	Edwards, 1924	28
C. scutellata	*Corynoneura scutellata*	Winnertz, 1846	118
S. bausei	*Stempellina bausei*	(Kieffer, 1911)	45
T. brundini	*Tanytarsus brundini*	Lindeberg, 1963	44
T. gregarius	*Tanytarsus gregarius*	Kieffer, 1909	123
T. volgensis	*Tanytarsus volgensis*	Miseiko, 1967	58
V. albisutus	*Virgatanytarsus albisutus*	(Santos-Abreu, 1918)	53
C. atridorsum	*Cladotanytarsus atridorsum*	Kieffer, 1924	100
C. mancus	*Cladotanytarsus mancus*	(Walker, 1856)	32
R. rhenanus	*Rheotanytarsus rhenanus*	Klink, 1983	74
P. austriacus	*Paratanytarsus austriacus*	(Kieffer, 1924)	26
P. dissimilis	*Paratanytarsus dissimilis*	(Johannsen, 1905)	88
P. lauterborni	*Paratanytarsus lauterborni*	(Kieffer, 1909)	62
P. mediterraneus	*Paratanytarsus mediterraneus*	Reiss & Säwedal, 1981	60
M. atrofasciata	*Micropsectra atrofasciata*	(Kieffer, 1911)	395
M. apposita	*Micropsectra apposita*	(Walker, 1856)	15
M. notescens	*Micropsectra notescens*	(Walker, 1856)	45
P. prasinatus	*Pseudochironomus prasinatus*	(Stäger, 1839)	62
P. albimanus	*Paratendipes albimanus*	(Meigen, 1818)	143
M. pedellus	*Microtendipes pedellus*	(De Geer, 1776)	205
P. orophila	*Pagastiella orophila*	(Edwards, 1929)	35
P. flavipes	*Phaenopsectra flavipes*	(Meigen, 1818)	134
P. P. sordens	*Polypedilum (Pentapedilum) sordens*	(van der Wulp, 1874)	81
P. P. laetum	*Polypedilum (Polypedilum) laetum*	(Meigen, 1818)	171
P. P. nubeculosum	*Polypedilum (Polypedilum) nubeculosum*	(Meigen, 1804)	320
P. T. scalaenum	*Polypedilum (Tripodura) scalaenum*	(Schrank, 1803)	34
P. U. convictum	*Polypedilum (Uresipedilum) convictum*	(Walker, 1856)	65
P. U. cultellatum	*Polypedilum (Uresipedilum) cultellatum*	Goetghebuer, 1931	70
E. tendens	*Endochironomus tendens*	(Fabricius, 1775)	89
C. C. anthracinus	*Chironomus (Chironomus) anthracinus*	Zetterstedt, 1860	101
C. C. riparius	*Chironomus (Chironomus) riparius*	Meigen, 1804	237
C. C. plumosus	*Chironomus (Chironomus) plumosus*	(Linnaeus, 1758)	196
D. nervosus	*Dicrotendipes nervosus*	(Stäger, 1839)	182
G. G. pallens	*Glyptotendipes (Glyptotendipes) pallens*	(Meigen, 1804)	106
C. viridulum	*Cladopelma viridulum*	(Linnaeus, 1767)	115
M. tener	*Microchironomus tener*	(Kieffer, 1918)	35
P. gracilior	*Parachironomus gracilior*	(Kieffer, 1918)	109
P. camptolabis	*Paracladopelma camptolabis*	(Kieffer, 1913)	57
P. nigritulum	*Paracladopelma nigritulum*	(Goetghebuer, 1942)	17
H. fuscimanus	*Harnischia fuscimanus*	(Kieffer, 1921)	58
C. defectus	*Cryptochironomus defectus*	(Kieffer, 1913)	156
D. vulneratus	*Demicryptochironomus vulneratus*	(Zetterstedt, 1838)	52

**Table 3 insects-15-00272-t003:** Results of partial canonical constrained ordination (pCCA). See Section 2 for abbreviations of environmental variables; spatial variables are a third-degree polynomial of longitude (=x) and latitude (=y) [25]. (**a**) pCCA with environmental variables as constraining factors and spatial variables as conditioning factors. The formula used was: species ~altit + dist + cond + O_2_ + temp + pH + TP + NH_4_ + Condition (x + x^2^ + x^3^ + y + xy + x^2^y + y^2^ + xy^2^ + y^3^); (**b**) inverse pCCA (pCCAi) with spatial variables as constraining factors and environmental variables as conditioning factors. The formula used was: species ~x + x^2^ + x^3^ + y + xy + x^2^y + y^2^ + xy^2^ + y^3^ + Condition (altit + dist + cond + O_2_ + temp + pH + TP + NH_4_).

(**a**) pCCA		
Partitioning of scaled chi-square	Inertia	Proportion
Total	7.5026	1.0000
Conditioned	0.3442	0.0459
Constrained	0.8090	0.1078
Unconstrained	6.3494	0.8463
(**b**) pCCAi		
Partitioning of scaled chi-square	Inertia	Proportion
Total	7.5026	1.0000
Conditioned	0.8742	0.1165
Constrained	0.2790	0.0372
Unconstrained	6.3494	0.8463

**Table 4 insects-15-00272-t004:** Results of partial canonical constrained ordination (pCCA) and its inverse (pCCAi). Eigenvalues of the first CCA axes, and their contribution to the scaled chi-square after removing the contribution of conditioning variables are shown.

(**a**) pCCA							
	CCA1	CCA2	CCA3	CCA4	CCA5	CCA6	CCA7
Eigenvalue	0.39553	0.18411	0.07284	0.05362	0.04247	0.02286	0.02073
Proportion explained	0.05525	0.02572	0.01018	0.00749	0.00593	0.00319	0.00290
Cumulative proportion	0.05525	0.08097	0.09115	0.09864	0.10457	0.10777	0.11066
(**b**) pCCAi							
	CCA1	CCA2	CCA3	CCA4	CCA5	CCA6	CCA7
Eigenvalue	0.07832	0.06361	0.05698	0.03346	0.02045	0.01039	0.00783
Proportion explained	0.01182	0.00960	0.00860	0.00505	0.00309	0.00157	0.00118
Cumulative proportion	0.01182	0.02141	0.03001	0.03506	0.03814	0.03971	0.04089

**Table 5 insects-15-00272-t005:** Number of sites originally present in each habitat and predicted in the same or in other habitats. The row sums are the total number of sites originally classified in one habitat; the column sums are the number of sites predicted in the same habitat.

Habitat	AL03	AL04	AL05	AL06	Alalp	Crenal	Kryal	ME04	ME07	Potamal	Rhithral	Original
AL03	60	0	0	0	0	0	0	1	0	0	7	68
AL04	0	6	1	0	0	0	0	0	0	0	1	8
AL05	3	1	38	2	0	0	0	1	0	1	7	53
AL06	8	1	4	38	0	0	0	1	0	0	1	53
AlAlps	0	0	0	1	17	0	0	0	0	0	0	18
Crenal	0	0	0	0	0	29	0	0	0	6	10	45
Kryal	0	0	0	0	2	0	60	0	0	0	8	70
ME04	0	0	0	0	0	1	0	17	0	0	4	22
ME07	0	0	0	0	0	0	0	0	15	0	0	15
Potamal	0	0	1	0	0	1	0	1	0	148	26	177
Rhithral	0	0	0	0	0	5	1	4	0	5	244	259
Predicted	71	8	44	41	19	36	61	25	15	160	308	

**Table 6 insects-15-00272-t006:** Moran’s eigenvector maps (MEMs), ordered according to R^2^ correlation with species matrix using *listw.candidates* and *listw.select* functions. Only the ten most significant vectors are given. The complete matrix is provided in Appendix A.

Order	Variables	R^2^	R^2^Cum	AdjR^2^Cum	*p*-Value
1	MEM1	0.0271	0.0271	0.0234	0.01
2	MEM5	0.0255	0.0526	0.0453	0.01
3	MEM23	0.0190	0.0716	0.0609	0.01
4	MEM16	0.0183	0.0899	0.0758	0.01
5	MEM19	0.0172	0.1071	0.0898	0.01
6	MEM3	0.0164	0.1235	0.1031	0.01
7	MEM4	0.0159	0.1395	0.1160	0.01
8	MEM2	0.0157	0.1552	0.1287	0.01
9	MEM13	0.0128	0.1679	0.1385	0.01
10	MEM29	0.0113	0.1792	0.1468	0.01

## Data Availability

All specimens examined and the full data matrix used in data analysis can be provided if requested from the first author.

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
