# Peer review of "Response of Chironomids (Diptera, Chironomidae) to Environmental Factors at Different Spatial Scales"

_insects, 2024, doi:10.3390/insects15040272_

Round 1
Reviewer 1 Report
Comments and Suggestions for Authors
Manuscript “Response of Chironomids (Diptera, Chironomidae) to Environmental Factors at Different Spatial Scales” by Bruno Rossaro and Laura Marziali.
GENERAL COMMENT
The paper presents a comprehensive review of factors influencing the distribution of chironomid species, focusing on spatial scales considering latitude, longitude and altitude, and water quality parameters. The authors analyse data from Italy and other countries, utilizing multivariate methods to examine chironomid species abundances across different water bodies. One key finding is the significance of habitat type in shaping chironomid assemblages. Overall, the paper provides valuable insights into the complex factors driving chironomid species distribution.
The work is well-structured and well-written, in the section below I highlight a few comments/corrections aimed at improving the clarity of the manuscript.
SPECIFIC COMMENTS
- Please correct the corresponding author's indication; one author is listed, but the associated email belongs to the other author.
Simple Summary
- line 10-11: “Chironomids are probably the richest species family among aquatic insects colonizing almost all freshwater habitats." The sentence "richest species family" is unclear. Perhaps you mean "most diverse family".
- lines 11-12: “The focus of the present research is the emphasis on taxonomic composition as the most efficient tool to describe the biodiversity of natural habitats." This sentence could be rephrased for clarity: "The emphasis of the present research is on taxonomic composition as the most efficient tool for describing the biodiversity of natural habitats.”
Abstract
- line 21: replace “to” with “on.
- lines 27-29: rewrite as follows “…The results confirmed previous conclusions, namely that habitat type, including different running waters reaches (kryal, krenal, rhthral, potamal) and different lake types, is the most significant factor…”
1. Introduction
- lines 35-36: “…often contributing to the highest observed diversity in the sample…” this sentence is unclear, clarify.
2. Materials and Methods
- line 82: how was the standardisation of environmental variables made?
3. Results
- lines 148-149: this statement is inaccurate, the high altitude alpine lakes are not represented in the graph to the right of the Kryal sites.
4. Discussion
- lines 226-229: I suggest that the sentence “Feld & Hering … Poland)” should be moved after, together with the discussion that starts at line 236.
- lines 262-265: these two sentences are unclear, clarify.
Tables
- Table 1 should be improved by standardizing the description for all habitat types, for instance, Krenal: springs, Kryal: glacial streams or streams in glacial areas.
- rewrite the caption of Table 1.
- rewrite the caption in Table 3, giving the full description (a and b) at the beginning.
- Table 4 (line 168) is erroneously indicated under Table 3.
Figures
- The figure should be moved forward in the results (after Table 3).- Figure 4 should be inserted before Table 4.
- Figure 8 should be inserted before Table 5.
Supplementary Material
Unfortunately, I could not see the supplementary material because it was not included for the review; however, I think that this did not affect substantially the revision process.
Author Response
Many thanks to reviewer 1; your review was very accurate and allowed us to ameliorate the manuscript; I accepted almost all your suggestions; in italic my answers to comments; I have corrected the text as suggested, unless specified in this rebuttal letter.
Comments and Suggestions for Authors
Manuscript “Response of Chironomids (Diptera, Chironomidae) to Environmental Factors at Different Spatial Scales” by Bruno Rossaro and Laura Marziali.
GENERAL COMMENT
The paper presents a comprehensive review of factors influencing the distribution of chironomid species, focusing on spatial scales considering latitude, longitude and altitude, and water quality parameters. The authors analyse data from Italy and other countries, utilizing multivariate methods to examine chironomid species abundances across different water bodies. One key finding is the significance of habitat type in shaping chironomid assemblages. Overall, the paper provides valuable insights into the complex factors driving chironomid species distribution.
The work is well-structured and well-written, in the section below I highlight a few comments/corrections aimed at improving the clarity of the manuscript.
SPECIFIC COMMENTS
- Please correct the corresponding author's indication; one author is listed, but the associated email belongs to the other author.
Corrected as suggested, see revised version
Simple Summary
- line 10-11: “Chironomids are probably the richest species family among aquatic insects colonizing almost all freshwater habitats." The sentence "richest species family" is unclear. Perhaps you mean "most diverse family".
Corrected as suggested, see revised version
- lines 11-12: “The focus of the present research is the emphasis on taxonomic composition as the most efficient tool to describe the biodiversity of natural habitats." This sentence could be rephrased for clarity: "The emphasis of the present research is on taxonomic composition as the most efficient tool for describing the biodiversity of natural habitats.”
Corrected as suggested, see revised version
Abstract
- line 21: replace “to” with “on.
Corrected as suggested, see revised version
- lines 27-29: rewrite as follows “…The results confirmed previous conclusions, namely that habitat type, including different running waters reaches (kryal, krenal, rhthral, potamal) and different lake types, is the most significant factor…”
Corrected as suggested, see revised version
- Introduction
- lines 35-36: “…often contributing to the highest observed diversity in the sample…” this sentence is unclear, clarify.
We have tried to correct, see revised version
- Materials and Methods
- line 82: how was the standardisation of environmental variables made?
See line 88-89 of the revised version
- Results
- lines 148-149: this statement is inaccurate, the high altitude alpine lakes are not represented in the graph to the right of the Kryal sites.
Now row 166-167 corrected as: The alpine lakes at high altitude were plotted near the kryal sites.
- Discussion
- lines 226-229: I suggest that the sentence “Feld & Hering … Poland)” should be moved after, together with the discussion that starts at line 236
- lines 262-265: these two sentences are unclear, clarify.
It is true, The discussion was deeply revised, see the corrected version
Tables
- Table 1 should be improved by standardizing the description for all habitat types, for instance, Krenal: springs, Kryal: glacial streams or streams in glacial areas.
- rewrite the caption of Table 1.
It is true, Table 1 was corrected as suggested
- rewrite the caption in Table 3, giving the full description (a and b) at the beginning.
Table 3 caption was corrected
- Table 4 (line 168) is erroneously indicated under Table 3.
OK corrected
Figures
- The figure should be moved forward in the results (after Table 3).- Figure 4 should be inserted before Table 4.
- Figure 8 should be inserted before Table 5.
I included Figures and Tables at the end of the text; the final position of Figures and Tables will be modified in agreement with the Editor
Supplementary Material
Unfortunately, I could not see the supplementary material because it was not included for the review; however, I think that this did not affect substantially the revision process.
Sorry but I have uploaded on the web site
Reviewer 2 Report
Comments and Suggestions for Authors
This is a very interesting article on the response of chironomids to environmental factors at different spatial scales. The article is well written but the authors have not given clear conclusions. The discussion should be restructured and rephrased to make it clearer what it is about, the author's findings or the findings from the papers cited by the authors. It would be nice if you could emphasize the conclusions and state whether the hypotheses tested are accepted or not. I think the manuscript can be published after minor corrections.
Line 63: “The input data matrix included 788 samples collected in 11 different habitats….” Then in line 74: “Raw data included 19531 samples and 255 variables. Only sites including values of at least 10 variables and variables present in at least 100 samples were retained. This selection left 782 sites…”.
It is not clear whether these are sites or samples? In Table 1 you have the number of sites? Please clarify and check throughout the text.
Line 71: Please capitalise the titles of tables and figures
Line 74: 255 variables – you have not specified what the variables include
Line 74: “Only sites including values….” Did you mean “Only samples including values…”
Line 111: Please insert a legend for the symbols and capitalise the first word of the picture title throughout text
Line 140: This line is unclear, does it belong here. Please check.
Line 160: cca replase with CCA
Line 162: “latitude (Y axis) and longitude (X axis)” did you mean vector instead of axis?
Line 168: “Table 3” please correct to Table 4
Line 208: “Figure 9. RDA results of species table constrained by spatial variables.” Maybe it is better to say, “RDA results of species matrix constrained by spatial variables.”
Line 236: “A comparison of our results with other studies having analysis of spatial factors as 236 the main objective [9] are not easy because of 3 important differences: …“ Why did you list 4?
Line 245: Why did you not use functional variables?
Line 252: “It is evident from the Moran’s maps that there is a spatial separation of species according to latitude and longitude, but it is surely an indirect effect, because the sites at highest altitudes are more frequent in the west part of the sampled area (West Alps), while the separation according to latitude is clearly bound to the fact the sites at the lower latitude have higher temperatures than the northern sites and effect of temperature on chironomid distribution is well documented [35] [36]. The factors more responsible of chironomids species distribution are confirmed to be a joint combination of substrate, water temperature, conductivity, current velocity, dissolved oxygen[2] [8] effects, which can be efficiently described using the terms kryal, krenal, rhithral, potamal, lentic or lotic habitat, clearing summarizing preferences to different factors interacting to each other.”
The sentence is confusing, it is not clear which refers to the results of this research and which to the earlier studies. I suggest correcting it to make it clearer.
I suggest: “It is evident from the Moran’s maps that there was a spatial separation of species according to latitude and longitude, but it was surely an indirect effect, because the sites at highest altitudes were more frequent in the west part of the sampled area (West Alps). The separation according to latitude was clearly bound to the fact the sites at the lower latitude have higher temperatures than the northern sites and effect of temperature on chironomid distribution is well documented [35] [36]. The factors more responsible of chironomids species distribution are confirmed to be a joint combination of substrate, water temperature, conductivity, current velocity, dissolved oxygen[2] [8] effects, which can be efficiently described using the terms kryal, krenal, rhithral, potamal, lentic or lotic habitat, clearing summarizing preferences to different factors interacting to each other.”
Line 247: This sentence is unclear rearrange it please: Feld & Hering [9] concluded that interactions at different spatial scales confounded the interpretation of the results, but in any case, the range scales (meso-scale) were the variables that were the greatest source of variation. Despite the different experimental designs of the two studies, the small spatial scale variables confirmed their predominant effect in both cases.
Author Response
Many thanks to reviewer 2; in italic my answers to his comments; I have corrected the text as suggested, unless specified in this letter.
Comments and Suggestions for Authors
This is a very interesting article on the response of chironomids to environmental factors at different spatial scales. The article is well written but the authors have not given clear conclusions. The discussion should be restructured and rephrased to make it clearer what it is about, the author's findings or the findings from the papers cited by the authors. It would be nice if you could emphasize the conclusions and state whether the hypotheses tested are accepted or not. I think the manuscript can be published after minor corrections.
OK Discussion was deeply revised, see the corrected version
Line 63: “The input data matrix included 788 samples collected in 11 different habitats….” Then in line 74: “Raw data included 19531 samples and 255 variables. Only sites including values of at least 10 variables and variables present in at least 100 samples were retained. This selection left 782 sites…”.
It is not clear whether these are sites or samples? In Table 1 you have the number of sites? Please clarify and check throughout the text.
It is true I have corrected samples with sites, site is true, sample is wrong
Line 71: Please capitalise the titles of tables and figures
Corrected as suggested
Line 74: 255 variables – you have not specified what the variables include
Added as suggested
Line 74: “Only sites including values….” Did you mean “Only samples including values…”
Corrected as suggested, in this case sample is true
Line 111: Please insert a legend for the symbols and capitalise the first word of the picture title throughout text
Corrected as suggested
Line 140: This line is unclear, does it belong here. Please check.
Modified, see revised text, now at row 160
Line 160: cca replase with CCA
Modified, see revised text, now at row 179
Line 162: “latitude (Y axis) and longitude (X axis)” did you mean vector instead of axis?
Modified, see revised text, now at row 181
Line 168: “Table 3” please correct to Table 4
Corrected as suggested
Line 208: “Figure 9. RDA results of species table constrained by spatial variables.” Maybe it is better to say, “RDA results of species matrix constrained by spatial variables.”
Corrected as suggested
Line 236: “A comparison of our results with other studies having analysis of spatial factors as 236 the main objective [9] are not easy because of 3 important differences: …“ Why did you list 4?
Corrected as suggested
Line 245: Why did you not use functional variables?
There are many reasons, the most relevant is my personal preference for species identification (taxonomy); I fear that feeding groups are not well established for chironomids; many species are opportunistic, so it is difficult to assign them to a feeding group
Line 252: “It is evident from the Moran’s maps that there is a spatial separation of species according to latitude and longitude, but it is surely an indirect effect, because the sites at highest altitudes are more frequent in the west part of the sampled area (West Alps), while the separation according to latitude is clearly bound to the fact the sites at the lower latitude have higher temperatures than the northern sites and effect of temperature on chironomid distribution is well documented [35] [36]. The factors more responsible of chironomids species distribution are confirmed to be a joint combination of substrate, water temperature, conductivity, current velocity, dissolved oxygen[2] [8] effects, which can be efficiently described using the terms kryal, krenal, rhithral, potamal, lentic or lotic habitat, clearing summarizing preferences to different factors interacting to each other.”
The sentence is confusing, it is not clear which refers to the results of this research and which to the earlier studies. I suggest correcting it to make it clearer.
I suggest: “It is evident from the Moran’s maps that there was a spatial separation of species according to latitude and longitude, but it was surely an indirect effect, because the sites at highest altitudes were more frequent in the west part of the sampled area (West Alps). The separation according to latitude was clearly bound to the fact the sites at the lower latitude have higher temperatures than the northern sites and effect of temperature on chironomid distribution is well documented [35] [36]. The factors more responsible of chironomids species distribution are confirmed to be a joint combination of substrate, water temperature, conductivity, current velocity, dissolved oxygen[2] [8] effects, which can be efficiently described using the terms kryal, krenal, rhithral, potamal, lentic or lotic habitat, clearing summarizing preferences to different factors interacting to each other.”
I completely agree with your suggestion, I have included in the text
Line 247: This sentence is unclear rearrange it please: Feld & Hering [9] concluded that interactions at different spatial scales confounded the interpretation of the results, but in any case, the range scales (meso-scale) were the variables that were the greatest source of variation. Despite the different experimental designs of the two studies, the small spatial scale variables confirmed their predominant effect in both cases.
I completely agree with your suggestion, I have included in the text
Reviewer 3 Report
Comments and Suggestions for Authors
I was asked to stop reviewing this paper by the Editor because two reviews already had been submitted. Following are my comments up to Methods. It is very difficult to write a scientific paper in a language that is not native to you. I have offered you some suggestions for improving the wording and flow of the text. The subject is interesting and it appears the authors have some compelling data.
Review of: Response of Chironomids (Diptera, Chironomidae) to Environmental Factors at Different Spatial Scales. By Bruno Rossaro and Laura Marziali
Simple summary
Line 10: Chironomids are probably the most speciose family among aquatic insects.
Line 12: In this paper, it is shown that the natural habitat type…
Line 14-15: This is because the habitat type best characterizes contributing environmental parameters such as substrate size and composition, water temperature….current velocity, and dissolved oxygen content.
Line 15-16: Different spatial scales are often proposed as relevant factors for modeling community composition, but …
Line 17: …only at small spatial scales, and environmental factors are more important determinants of species distributions.
Abstract
Lines 19-20: Factors responsible for species distributions, including responses at different spatial scales, have been previously investigated.
Line 21: ‘…factors explain the distributions of chironomid species focusing on factors operating….regional, and waterbasin level variables.’ [country is a political descriptor, and not an natural entity. That reference should be deleted.]
Line 24-25. …were analyzed using partial canonical correspondence analysis and multiple discriminant analysis.
Line 27-30: The results showed that habitat type, including different lotic waters (i.e., kryal, krenal, rhthral, potamal) and different lake types, is the most significant factor in separating chironomid assemblages, while spatial factors acted only as indirect influences.
Introduction
[The first two paragraphs are a single sentence each. Combine them and re-write as suggested below]
Line 34-50: Chironimids are a speciose aquatic insect family with about 15,000 described species, and often contributing a large portion of the insect diversity in the benthos [1]. Their ecological tolerances also are diverse with many species being tolerant to extreme environmental conditions [1]. The various species have been shown to respond to a broad suite of different environmental (=abiotic) factors, such as water temperature, salinity, and sediment composition [2], but biotic factors (competition, predation) are also important [3]. Furthermore, interactions between different combinations of abiotic and biotic factors as well as spatial and temporal contiguities are complex and make the species distributions difficult to interpret. Such complexities are further exacerbated in environmentally unstable aquatic habitats such as those in the Mediterranean, which are known to broadly fluctuate among years [5]. Those habitats will likely become more unstable do to global climate change [5].
Historical biogeographic factors are known to influence chironomid species composition in aquatic habitats at large spatial scales [6], but in more specific habitats such as headwater streams, it can be supposed those factors act at a smaller spatial scale [7]. Despite this described complexity, often only a few key factors may account for the largest portion of observed variation [8]. The scarcity of samples and data available for study coupled with different sampling methods and tools and differing taxonomic levels used (i.e., family, genus, species), produces further uncertainty in interpreting results. Few previous studies have evaluated the responses of chironomids at different spatial scales [9,10], and they primarily focused on small spatial scales [11. 12].
[your last paragraph contains some methods content and should be moved there. I suggest the following:]
The aim of the present study is to assess the importance of environmental factors acting at different spatial scales on chironomid communities using Moran’s eigenvector maps (MEM) [13].
Materials and Methods
[Review stopped here at request of Editor]
Comments on the Quality of English Language
The writing, especially the flow, needs to be improved.
Author Response
Many tanks to reviewer 3.
Thank you for having suggested very appropriate corrections in the text, I have included them in the final manuscript
Comments and Suggestions for Authors
I was asked to stop reviewing this paper by the Editor because two reviews already had been submitted. Following are my comments up to Methods. It is very difficult to write a scientific paper in a language that is not native to you. I have offered you some suggestions for improving the wording and flow of the text. The subject is interesting and it appears the authors have some compelling data.
Review of: Response of Chironomids (Diptera, Chironomidae) to Environmental Factors at Different Spatial Scales. By Bruno Rossaro and Laura Marziali
Simple summary
Line 10: Chironomids are probably the most speciose family among aquatic insects.
corrected
Line 12: In this paper, it is shown that the natural habitat type…
Line 14-15: This is because the habitat type best characterizes contributing environmental parameters such as substrate size and composition, water temperature….current velocity, and dissolved oxygen content.
Line 15-16: Different spatial scales are often proposed as relevant factors for modeling community composition, but …
Line 17: …only at small spatial scales, and environmental factors are more important determinants of species distributions.
Abstract
Lines 19-20: Factors responsible for species distributions, including responses at different spatial scales, have been previously investigated.
Line 21: ‘…factors explain the distributions of chironomid species focusing on factors operating….regional, and waterbasin level variables.’ [country is a political descriptor, and not an natural entity. That reference should be deleted.]
Line 24-25. …were analyzed using partial canonical correspondence analysis and multiple discriminant analysis.
Line 27-30: The results showed that habitat type, including different lotic waters (i.e., kryal, krenal, rhthral, potamal) and different lake types, is the most significant factor in separating chironomid assemblages, while spatial factors acted only as indirect influences.
Introduction
[The first two paragraphs are a single sentence each. Combine them and re-write as suggested below]
Line 34-50: Chironomids are a speciose aquatic insect family with about 15,000 described species, and often contributing a large portion of the insect diversity in the benthos [1]. Their ecological tolerances also are diverse with many species being tolerant to extreme environmental conditions [1]. The various species have been shown to respond to a broad suite of different environmental (=abiotic) factors, such as water temperature, salinity, and sediment composition [2], but biotic factors (competition, predation) are also important [3]. Furthermore, interactions between different combinations of abiotic and biotic factors as well as spatial and temporal contiguities are complex and make the species distributions difficult to interpret. Such complexities are further exacerbated in environmentally unstable aquatic habitats such as those in the Mediterranean, which are known to broadly fluctuate among years [5]. Those habitats will likely become more unstable do to global climate change [5].
Historical biogeographic factors are known to influence chironomid species composition in aquatic habitats at large spatial scales [6], but in more specific habitats such as headwater streams, it can be supposed those factors act at a smaller spatial scale [7]. Despite this described complexity, often only a few key factors may account for the largest portion of observed variation [8]. The scarcity of samples and data available for study coupled with different sampling methods and tools and differing taxonomic levels used (i.e., family, genus, species), produces further uncertainty in interpreting results. Few previous studies have evaluated the responses of chironomids at different spatial scales [9,10], and they primarily focused on small spatial scales [11. 12].
[your last paragraph contains some methods content and should be moved there. I suggest the following:]
The aim of the present study is to assess the importance of environmental factors acting at different spatial scales on chironomid communities using Moran’s eigenvector maps (MEM) [13].
Materials and Methods
[Review stopped here at request of Editor]
Reviewer 4 Report
Comments and Suggestions for Authors
The study of Rossaro and Marziali investigated differences in chironomid assemblages resulting from differences in several environmental variables at different spatial scales. The study was focused on the species level, which is very good, considering the lack of ecological studies on those aquatic insects. However, in my opinion, the manuscript needs to be significantly improved and better written before it could be suitable for publication. I would advise authors to spend some more time in improving each section of the manuscript, to make some parts more advanced or clearer.
The language should be improved, on some occasions, it was not clear what authors wanted to say – so I would suggest a native speaker reviews the manuscript.
The introduction is quite short, the authors could add more information about Chironomidae in general, their ecological requirements, such as habitat selection, life history traits etc. Also, they should highlight why is their research important – it is well known that assemblages of various aquatic insects differ among different habitat types – why is it important to study that in Chironomidae (e.g. lack of ecological, distributional data etc.).
Methods – the authors should add what analysis they used to assess particular goal.
Results – the authors should add description of all results, not just refer us to particular table or figure. Also, no need to discuss results there, but in the Discussion section.
The discussion could be improved – now it is a mix of results/methods repetitions, sections with only literature data that are not well connected with results. Much better way would be present the results obtained within the current study, and then comment those results (discuss them) with help of literature.
The conclusions should be expanded, and more highlight what novel and most important finding of this study is, because in the current form, it is not completely clear.
More detailed comments can be found within the manuscript pdf file.

The language should be improved, on some occasions, it was not clear what authors wanted to say – so I would suggest a native speaker reviews the manuscript.
Author Response
Referee 4 (N.B. my answers are in italic, original text in roman)
In italics my answers
Many thanks for the intensive valuable work, rich of very useful suggestions.
I have tried to adjust the text to all suggestions (see the revised text), about only few points I had some problem in following reviewer’s suggestion; I have exposed them below
The study of Rossaro and Marziali investigated differences in chironomid assemblages resulting from differences in several environmental variables at different spatial scales. The study was focused on the species level, which is very good, considering the lack of ecological studies on those aquatic insects. However, in my opinion, the manuscript needs to be significantly improved and better written before it could be suitable for publication. I would advise authors to spend some more time in improving each section of the manuscript, to make some parts more advanced or clearer.
The language should be improved, on some occasions, it was not clear what authors wanted to say – so I would suggest a native speaker reviews the manuscript.
The introduction is quite short, the authors could add more information about Chironomidae in general, their ecological requirements, such as habitat selection, life history traits etc. Also, they should highlight why is their research important – it is well known that assemblages of various aquatic insects differ among different habitat types – why is it important to study that in Chironomidae (e.g. lack of ecological, distributional data etc.).
Introduction has been rewritten, but I think that general info about Chironomids is better found in the cited literature (Reference [1] in particular), and it is not necessary to summarize in this paper
Methods – the authors should add what analysis they used to assess particular goal.
The analyses used were added in the text, detailed description of methods is in cited literature (Reference [24,25,29] in particular)
Results – the authors should add description of all results, not just refer us to particular table or figure. Also, no need to discuss results there, but in the Discussion section.
I have tried to do, see revised text References and Discussion which was deeply revised
The discussion could be improved – now it is a mix of results/methods repetitions, sections with only literature data that are not well connected with results. Much better way would be present the results obtained within the current study, and then comment those results (discuss them) with help of literature.
Discussion which was revised
The conclusions should be expanded, and more highlight what novel and most important finding of this study is, because in the current form, it is not completely clear.
Conclusion paragraph was added
More detailed comments can be found within the manuscript pdf file.
I agree with all suggested/requested corrections made in pdf, and I have made the corrections in the docx file I have returned
I add the following observations to referee notes present in pdf
Reference [5] at line 45 is correct
References [15], [16] at line 65 are OK both papers describe the Classification proposed by Tartari G.
Table 1
AL04 does not include large lakes , no lake with Area > 100 km2 has a depth < 15 m
Comment at line 81 software used was added
Comment at line 106 we refer to web site https://cran.r-project.org/web/packages/adespatial/vignettes/tutorial.html (2023-10-18).
In this tutorial the point 6.3 is entitled Canonical analysis, but a redundancy analysis with species matrix as dependent variables and spatial matrix as constraining variables is performed; the author (Stephan Dray) seems to consider redundancy analysis a canonical analysis
Comment at line 159: if one enlarges the fonts the names will become overlapped
Comment at line 205 fig 8: if one enlarges points they will become overlapped, it is not necessary to see each point, different colors can give an idea of the distribution of points
Comment at line 264: anthropogenic stress was not considered in the present analysis, my citation refers to literature data [2, 8, 18]
Comment at line 226: the aim of this paragraph is to compare our research with other papers, underlining that it is very difficult to do a comparison
Comment at line 238: I have just cited the authors, I think it is not necessary to repeat it each time
Comment at line 286: you confirm me that spatial factors do not act as such, but mediated by environmental factors, so I think to leave the sentence unchanged
Round 2
Reviewer 3 Report
Comments and Suggestions for Authors
The manuscript through the introduction is much improved. However, my review stopped at the introduction and did not include the remainder of the manuscript. I therefore am unable to make a judgement call on acceptability.
Author Response
No modification was suggested by reviewer 3 in the second review round.
Thank you again to reviewer 3 for his very accurate review and suggestions in the part of the work he analyzed.
Reviewer 4 Report
Comments and Suggestions for Authors
The authors have improved their manuscript, however, there are still points that need to be addressed before it could be considered for publication. In my previous review, I have suggested the language and style revision from a native speaker, which was not done, and as it is now, the text has many flaws. Also, I still find the results section as not sufficiently well written – as I said last time, it is not enough to refer to a particular table, or appendix (or figure), the authors also need to write the most important findings included in those tables (and figures) also within the text. I have highlighted that in the text several times, but this applies for all results which were not presented in the recommended manner. Also, the discussion is still a bit chaotic, made of so many small paragraphs, some seam to repeat more that once, while can be combined to one larger paragraph. The authors should first present their results and then discuss them with help of literature cited – I gave some suggestions in the MS text.

Needs to be revised by a native speaker.
Author Response
We thank the reviewer for his very accurate review.
In roman the reviewer suggestions.
In Italics our answers to still open questions.
The authors have improved their manuscript, however, there are still points that need to be addressed before it could be considered for publication.
Thank you for your suggestions, we have corrected the critical points trying to follow all your suggestions. The manuscript is now deeply modified trying to fulfill all reviewer’s requests.
In my previous review, I have suggested the language and style revision from a native speaker, which was not done, and as it is now, the text has many flaws.
Now the manuscript was reviewed by a researcher working in Ireland, who just successfully reviewed previous manuscripts
Also, I still find the results section as not sufficiently well written – as I said last time, it is not enough to refer to a particular table, or appendix (or figure), the authors also need to write the most important findings included in those tables (and figures) also within the text. I have highlighted that in the text several times, but this applies for all results which were not presented in the recommended manner.
In Results we have extended the text adding more explanations of all figures and tables. We have joined Figures 1-4 and 6-7 to better balance text with figures. Figure 8 was redrawn replacing the small points with ones of larger size. Figure 9 was really difficult to be read so we have replaced it with two larger figures. As a consequence the numbers of figures are modified, figs1,2,3,4 become fig 1a,b,c,d, fig 5 now is fig 2, figs 6, 7 now is fig 3a,b, fig 8 now is fig 4, fig 9 now become figs 5,6.
We agree that the methods used are difficult for non specialists, but the potential readers are invited to consult the original papers explaining the methods better then us
Also, the discussion is still a bit chaotic, made of so many small paragraphs, some seam to repeat more that once, while can be combined to one larger paragraph. The authors should first present their results and then discuss them with help of literature cited – I gave some suggestions in the MS text.
The Discussion is deeply emended, completely rewritten, discussion of our results is now well separated by discussion of literature cited, results obtained with the different methods (pCCA, pCCAi, DISCR, Moran and Mantel autocorrelation, Moran’s Eigenvector Maps) are discussed in detail, a Conclusion paragraph was added